# The Palaeoclimate and Terrestrial Exoplanet Radiative Transfer Model Intercomparison Project (PALAEOTRIP): experimental design and protocols

Colin Goldblatt, Lucas Kavanagh, and Maura Dewey

School of Earth and Ocean Sceinces, University of Victoria, Victoria, BC, Canada.

*Correspondence to:* Colin Goldblatt (czg@uvic.ca and info@palaeotrip.org)

**Abstract.** Accurate radiative transfer calculation is fundamental to all climate modelling. For deep palaeoclimate, and increasingly terrestrial exoplanet climate science, this brings both the joy and the challenge of exotic atmospheric compositions. The challenge here is that most standard radiation codes for climate modelling have been developed for modern atmospheric conditions, and may perform poorly away from these. The palaeoclimate or exoclimate modeller must either rely on these or use bespoke radiation codes, and in both cases rely on either blind faith or *ad hoc* testing of the code. In this paper, we describe the protocols for the Palaeoclimate and Terrestrial Exoplanet Radiative Transfer Model Intercomparison Project (PALAEOTRIP) to systematically address this. This will compare as many radiation codes used for palaeoclimate or exoplanets as possible, with the aim to identify the ranges of far-from-modern atmospheric compositions in which the codes perform well. This paper describes the experimental protocol and invites community participation in the project through 2017–2018.

## 1 Introduction

Earth's atmospheric composition has varied dramatically through time, and yet-to-be-discovered terrestrial exoplanets will add untold diversity. A example model of late Archean atmospheric composition would be of 30,000 ppmv $CO_2$, 1000 ppmv $CH_4$, no oxygen or ozone and an unknown nitrogen inventory, whereas escape from 'snowball Earth' glaciation may take 10% $CO_2$. A fundamental part of the palaeoclimate problem, and equivalently the exo-climate problem, may be stated as: given some atmospheric composition, what was the energy balance of the planet? Or, for given atmospheric composition and incident solar flux, what was the surface temperature?

To first approximation, an atmospheric composition and structure needs to be given, surface properties specified, then the radiation field must be simulated, the equations for which are well known (e.g. Goody and Yung, 1989). Regrettably, implementation is far from simple. Millions of gas absorption lines from numerous gases are relevant to the climate problem. Herculean work has assembled most of these into large and oft-revised databases (e.g. Rothman et al., 2013). From these databases, absorption cross sections may be calculated as a function of temperature and pressure. Even these cross sections, calculated with standard assumptions regarding the shape of absorption lines, have some notable disagreement with observations and smoothly varying "continuum" absorption must be added to produce realistic cross sections. Armed with cross sections, the radiative transfer equations may then be solved at the natural resolution of the lines—a so-called line-by-line calculation. Alas, these

can take time on the order of minutes to hours for a single column, hence are too slow by many orders of magnitude to be used in a climate model.

In a general circulation model (GCM), the radiative transfer for a single column must be evaluated in a fraction of a second. Consequently, simplifications must be made in the treatment of the radiative transfer and the spectral dependence must be heavily parameterized. To optimize efficiency, these parameterizations may be made for limited ranges of atmospheric composition or column abundances of absorbing molecules. Often, these parameterizations were made a decade or more ago, with poor documentation. Where an older (and likely faster) GCM is used for palaeoclimate research, one is automatically in the situation of using a legacy radiation code.

At the other end of the modelling spectrum, there still exists a cottage industry of bespoke development of fast-enough radiative transfer codes for deep palaeoclimate, planetary atmospheres, or other obscure radiative transfer problems, where all the required steps are made ad hoc. However, in some cases, the resources required to sufficiently test the code are unavailable locally.

Three broad classes of problem arise. First, whilst excellent parametrization is possible within design ranges, some parameterizations do not perform as well as a third-party user may hope. For example, intercomparison of radiation codes used for the IPCC Forth Assessment Report (Collins et al., 2006) showed that many codes simulated the changes due to a doubling of carbon dioxide poorly. Second, performance of codes will decrease outside design ranges, which often includes the regions that we are interested in for palaeoclimate (e.g. Goldblatt et al., 2009b). Third, errors are made in parameterizations (especially in bespoke codes) which can remain undetected through review and for some years afterwards.

The palaeoclimate or exoplanet modeller is thus in a bind. The science interest is in novel atmospheric compositions, whose radiation properties are outside the intuition of most non-specialists. It would be prudent to test *any* fast radiation code that one planned to use against a well-trusted line-by-line code across the parameter space of interest (e.g. Goldblatt et al., 2009b; Wolf, 2013; Yang et al., 2016); however, doing this requires both the specialist knowledge in radiative transfer, the local availability of such a model and a lot of time and energy. All of these can be hard to come by.

With the Palaeoclimate and Terrestrial Exoplanet Radiative Transfer Model Intercomparison Project (PALAEOTRIP), we hope to alleviate this problem. Our aim is to test a large number of fast radiation codes, both GCM and bespoke, against line-by-line models for a wide range of conditions applicable to palaeoclimate and terrestrial exoplanet research. Such intercomparison studies have a long history in application to modern conditions and anthropogenic global change (e.g. Ellingson et al., 1991; Fouquart et al., 1991; Collins et al., 2006; Oreopoulos et al., 2012; Pincus et al., 2015, 2016) and have contributed markedly to improvements in the fidelity of radiation codes and thus the robustness of climate models. Our hope is that by exporting such a systematic intercomparison process to deep palaeoclimate and exoplanets will yield similar improvements. In this paper, we describe the experimental design and protocol[1]. Up-to-date project information will be available at www.palaeotrip.org throughout the project.

---

[1]Community input on the experimental design and protocols was gathered during the open peer review process "discussion" phase of *Geoscientific Model Development*.

## 2 Experimental Design

### 2.1 Philosophy

Our hope it that by assembling and analyzing results from many radiative transfer codes outside of modern conditions, we will both help future investigators to make an educated choice of which radiative transfer code is applicable for a particular experiment, and inform model developers of opportunities for improvement of models.

The standard method of radiative transfer intercomparison is to compare model output—especially changes in fluxes in response to changes atmospheric composition—calculated on *fixed* atmospheric profiles. The use of fixed profiles is essential to isolate the fidelity of the radiative transfer codes (to be evaluated) from the myriad of other processes that determine the atmospheric profile. This methodology has a long history (e.g. Ellingson et al., 1991; Fouquart et al., 1991); see Collins et al. (2006) for an in-depth discussion of this methodology. We use instantaneous (unadjusted) radiative forcings. The most modern radiative transfer intercomparison project for IPCC class models (Pincus et al., 2016) additionally use *effective radiative forcings* that account for a variety of rapid adjustments in GCMs; these are not included here. Our method here corresponds to the (Pincus et al., 2016) assessment of "parametrization error".

Three groups of experiments are included, addressing changes to clear-sky properties under both a solar and a M-star spectrum, and adding clouds under the solar spectrum. These give fourteen experiments in total, each of which varies a parameter of key importance for palaeoclimate and Earth-like exoplanets. The choice of parameter space represents a range of mainstream assumptions about atmospheric composition through Earth history. We have explored all of this parameter space previously: see Goldblatt et al. (2009b) and Byrne and Goldblatt (2014a, b) for well mixed greenhouse gases, Goldblatt and Zahnle (2011b, a) for clouds and Goldblatt et al. (2009a) for varying atmospheric pressure. One class of model atmospheres that we exclude is $H_2$ dominated atmospheres (Wordsworth and Pierrehumbert, 2013), as air-broadened line shapes will likely not be appropriate for these, consequently a majority of codes may not perform well (that is, these atmospheres require rather specialist treatment, beyond the scope of this intercomparison).

Participating groups should run the experiments that their models are configured for, and omit any which are not possible (or onerous) to run. We do not expect groups do perform model development in order to participate in this project. For example, a model which had the solar spectrum hard coded and did not include $N_2O$ absorption would run experiments 1, 2a–b, 3–6 and 13–16. A model without clouds would omit experiments 13–16.

If some absorbing gases are missing, experiments which do not focus on these can still be run, with notes in the metadata and in discussion with the project team. As or analysis will focus on forcings (change from standard conditions) comparison to the standard conditions *from that model* will minimise the effect of any systematic offset from missing absorbers. For example, models without oxygen or ozone absorption could still run the experiments focussing on clouds.

If, for any reason, there is a limit to the number of experiments that a group can run then experiments 1–6 should be considered "core" and prioritized. A minimal set of experiments would be 1 and 2.

All of the required input files for the project are available at www.palaeotrip.org, and as an online supplement to this paper.

### 2.2 Model atmosphere

#### 2.2.1 Atmospheric profile

For simplicity, all experiments use a Global Annual Mean (GAM) profile. This based on a profile derived from averaging of reanalysis data by Byrne and Goldblatt (2014a). This specific profile should be used, and none substituted for it. We refer to model levels are the boundary between model layers. Experiments 1–4, 6–8 and 10 use the GAM profile unmodified, whereas experiments 5 and 9 modify it as described for experiment 5.

Radiatively active species in the atmosphere are $CO_2$, $CH_4$, $N_2O$, $H_2O$, $O_3$ and $O_2$. All mixing ratios are in parts per volume. Standard mixing ratios are 0.21 for $O_2$, and vertically resolved profiles supplied with in the GAM profile for $H_2O$ and $O_3$. For the remaining gases, referred to as well-mixed greenhouse gases (WMGHG), mixing rations are supplied in Table 1.

#### 2.2.2 Line data

Line-by-line codes should use line data from HITRAN2012 (e.g. Rothman et al., 2013).

Bespoke, GCM and legacy radiation codes will use a variety of line data. It is acceptable to submit either the most current/standard version, or a variety of versions corresponding to different applications. The model version number/name, a brief description and/or link to the full description should be included as metadata with the model output, especially the version number/name of the code.

#### 2.2.3 Stellar fluxes

Stellar fluxes are supplied for both the Sun and an example M-star (ADLeo) for models in which these are input directly.

As with line data, for codes which use a standard stellar flux, use this standard configuration and include whatever description possible. For such codes, where it is impractical to modify the stellar flux to an M-star, perform experiments 1–6 and 12–16 only.

All experiments should use an integrated stellar flux (solar constant) of $1360\,\mathrm{W\,m^{-2}}$.

#### 2.2.4 Clouds

Experiments with both low and high clouds are included. Calculations should be done with a single profile, with a cloud fraction of unity. Clouds may be specified in different ways in different radiation codes; the nominal descriptions here should be matched as well as possible given how clouds are specified in the particular radiation code, and appropriate description provided as metadata. We emphasize that the normal implementations of clouds in participant models should be used; single scattering properties are provided only for cases where this necessarily needs to be input. There a range of good choices of representation of cloud microphysics in models (i.e. which are different but entirely reasonable), so variation in the radiative effects of clouds may arise from these rather than error *per se*. Nonetheless, it is of primary interest to us how the radiative effects of clouds do vary when every attempt has been made to specify cloud physical properties equivalently.

Vertical position: if clouds are specified in a layer, low clouds should be in the 900–925 hPa layer, high clouds in the 250–300 hPa layer. If they are specified on levels, they should be at 912.5 hPa and 275 hPa, and can be specified with minimal vertical extent (or extent not exceeding the boundaries of the layer).

Low clouds are taken to be made of liquid water droplets. Thus cloud particles are well described as Mie spheres, so consistent specification across models should be straightforward. A standard low cloud should have a water path of $W = 40\,\mathrm{g\,m^{-2}}$ and effective radius of $10\,\mu\mathrm{m}$ (Goldblatt and Zahnle, 2011b). For codes which require single scattering properties, output from a Mie code is provided in the supplementary information (for simplicity, a single particle radius of $r = r_{\mathrm{eff}}$ is used) and the Henyey-Greenstein phase function should be used.

High clouds are taken to be made of ice crystals, and are thus more complicated to describe, as there are a variety of ice habits which are all non-spherical. The normal parameter to describe the size of particles is the effective diameter, $D_{\mathit{eff}}$. A standard high cloud should have a water path of $W = 20\,\mathrm{g\,m^{-2}}$ and effective diameter of $80\,\mu\mathrm{m}$ (Goldblatt and Zahnle, 2011b). For codes which require single scattering properties, these are taken from the "general habit mixture" of Baum et al. (2014, see also http://www.ssec.wisc.edu/ice_models/polarization.html) are provided in the supplementary information and the Henyey-Greenstein phase function should be used.

For codes which specify cloud thickness via an optical depth $\tau$, this can be calculated directly from the extinction efficiency, $Q$: $\tau = \pi r^2 n Q$ where $n$ is the number of cloud particles in the column. $n$ is found directly as $n = W/m$, where for liquid droplet mass is found $m = \rho(4/3)\pi r^3$, given density $\rho$ and for ice droplets $m$ is supplied.

### 2.2.5 Miscellaneous details

A solar zenith angle of $60°$ should be used for all experiments.

The surface should be black for thermal calculations and have a grey albedo of 0.12 in solar calculations. If a combined solar-thermal calculation is performed, the separation between solar and thermal albedos should be at $3\,\mu\mathrm{m}$.

The surface temperature is 288.24 K in all experiments.

Note that, for most experiments, a literal interpretation of the changes to atmospheric conditions will imply some physical inconsistencies: there is no change in atmospheric pressure when $CO_2$ mixing ratio increases to $10^{-1}$, water vapour may become super-saturated, there is no change to the T-p profile when gas concentrations change. These inconsistencies are tolerated, with the philosophy of designing simple and easy to compare experiments which test the fidelities of the radiation codes, which is best done on fixed profiles.

### 2.3 Experiments

The experiments are described in Table 1.

The runcode is a unique identifier for each run, which should be used as the name of the output file for each run (e.g. `runcode.dat`). These all begin `PT` (for palaeotrip, and to avoid starting a filename with a number), followed by the number of the experiment and the run number ($x$) within each experiment, counting from the lowest value of any quantity varied.

Table 1: Description of experiments

| Expt # | Parameter | Value/Desription |
|---|---|---|
| 1 | Name | Standard Conditions |
| | Description | - |
| | Spectrum | Solar |
| | Profile | GAM |
| | WHGHG | $400 \times 10^{-6}$ $CO_2$, $1 \times 10^{-6}$ $CH_4$, $1 \times 10^{-6}$ $N_2O$. |
| | Absorbers | $CO_2$, $CH_4$, $N_2O$, $H_2O$, $O_3$, $O_2$. |
| | Clouds | None |
| | Runcode | `PT1` |
| | Num. of runs | 1 |
| 2 | Name | WMGHG variation |
| | Description | The concentration of each WMGHG is varied in series (ranges below), with the other two held at standard conditions. The lower end of each range is selected for minimal radiative significance of that gas (see Byrne and Goldblatt, 2014b). The upper limit is an arbitrary guess at an upper bound for an Earth-like planet. Models should be run with concentrations evenly spaced in log units, with two runs per one log unit (e.g. $\{1 \times 10^{-9.0}, 1 \times 10^{-8.5}, 1 \times 10^{-8.0}, \dots \}$). |
| | Spectrum | Solar |
| | Profile | GAM |
| | WHGHG | (a) $CO_2$ from $10^{-9}$ to $10^{-1}$, $1 \times 10^{-6}$ $CH_4$, $1 \times 10^{-6}$ $N_2O$. |
| | | (b) $400 \times 10^{-6}$ $CO_2$, $CH_4$ from $10^{-9}$ to $10^{-2}$, $1 \times 10^{-6}$ $N_2O$. |
| | | (c) $400 \times 10^{-6}$ $CO_2$, $1 \times 10^{-6}$ $CH_4$, $N_2O$ from $10^{-9}$ to $10^{-2}$. |
| | Absorbers | $CO_2$, $CH_4$, $N_2O$, $H_2O$, $O_3$, $O_2$. |
| | Clouds | None |
| | Runcode | `PT2a_`*x*, `PT2b_`*x*, `PT2c_`*x* for $CO_2$, $CH_4$ and $N_2O$ respectively. |
| | Num. of runs | $17 + 15 + 15 = 47$ |
| 3 | Name | WMGHG variation, high background, anoxic. |
| | Description | The concentration of each WMGHG is varied in series, with the other two held at high conditions potentially representative of the Archean: $30,000 \times 10^{-6}$ $CO_2$, $300 \times 10^{-6}$ $CH_4$, $30 \times 10^{-6}$ $N_2O$. Absorption by atmospheric oxygen and ozone should be turned off, with all other conditions as standard. Note there is no change to the T-p profile. Otherwise, as experiment 2. |
| | Spectrum | Solar |
| | Profile | GAM |

*Continued on next page*

| Expt # | Parameter | Value/Desription |
|---|---|---|
| | WHGHG | (a) $CO_2$ from $10^{-9}$ to $10^{-1}$, $300 \times 10^{-6}$ $CH_4$, $30 \times 10^{-6}$ $N_2O$. |
| | | (b) $30,000 \times 10^{-6}$ $CO_2$, $CH_4$ from $10^{-9}$ to $10^{-2}$, $30 \times 10^{-6}$ $N_2O$. |
| | | (c) $30,000 \times 10^{-6}$ $CO_2$, $300 \times 10^{-6}$ $CH_4$,$N_2O$ from $10^{-9}$ to $10^{-2}$. |
| | Absorbers | $CO_2$, $CH_4$, $N_2O$, $H_2O$. |
| | Clouds | None |
| | Runcode | `PT3a_`*x*, `PT3b_`*x*, `PT3c_`*x* for $CO_2$, $CH_4$ and $N_2O$ respectively. |
| | Num. of runs | $17 + 15 + 15 = 47$ |
| 4 | Name | Water vapour variation |
| | Description | The water vapour mixing ratio is changed by a constant factor, with all other gases as standard conditions. The range of factors is $0.01 < x < 10$, which correspond to the differences a range of saturation vapour pressures from $230\,K$ to $330\,K$. Models should be run with concentrations evenly spaced in log units, with four runs per one log unit. |
| | Spectrum | Solar |
| | Profile | GAM, altered water vapour profiles |
| | WHGHG | $400 \times 10^{-6}$ $CO_2$, $1 \times 10^{-6}$ $CH_4$, $1 \times 10^{-6}$ $N_2O$. |
| | Absorbers | $CO_2$, $CH_4$, $N_2O$, $H_2O$, $O_3$, $O_2$. |
| | Clouds | None |
| | Runcode | `PT4_`*x* |
| | Num. of runs | 13 |
| 5 | Name | Surface pressure variation |
| | Description | The surface pressure is varied between 0.1 and 10 bars. This is done by multiplying the pressure vector in the GAM profile by a factor $0.1 \leq y \leq 10$, and dividing mixing ratio vectors of minor absorbing species ($CO_2$, $CH_4$, $N_2O$ and $O_3$) by $y$ so that the mass of each absorber is conserved. Absorption by atmospheric oxygen and ozone should be turned off, because the mass of this absorber cannot be conserved at low pressure. Models should be run with $y$ evenly spaced in log units, with four runs per one log unit. |
| | Spectrum | Solar |
| | Profile | GAM with modified pressure. |
| | WHGHG | $400 \times 10^{-6}$ $CO_2$, $1 \times 10^{-6}$ $CH_4$, $1 \times 10^{-6}$ $N_2O$. |
| | Absorbers | $CO_2$, $CH_4$, $N_2O$, $H_2O$. |
| | Clouds | None |
| | Runcode | `PT5_`*x* |

| Expt # | Parameter | Value/Desription |
|---|---|---|
| | Num. of runs | 9 |
| 6 | Name | No oxygen or ozone absorption |
| | Description | Absorption by atmospheric oxygen and ozone should be turned off, with all other conditions as standard. Note there is no change to the T-p profile. |
| | Spectrum | Solar |
| | Profile | GAM |
| | WHGHG | $400 \times 10^{-6}$ $CO_2$, $1 \times 10^{-6}$ $CH_4$, $1 \times 10^{-6}$ $N_2O$. |
| | Absorbers | $CO_2$, $CH_4$, $N_2O$, $H_2O$. |
| | Clouds | None |
| | Runcode | `PT6` |
| | Num. of runs | 1 |
| 7 | Name | Standard Conditions, M-star spectrum |
| | Description | As experiment 1, M-star spectrum substituted for solar spectrum. |
| | Spectrum | M-star |
| | Profile | GAM |
| | WHGHG | $400 \times 10^{-6}$ $CO_2$, $1 \times 10^{-6}$ $CH_4$, $1 \times 10^{-6}$ $N_2O$. |
| | Absorbers | $CO_2$, $CH_4$, $N_2O$, $H_2O$, $O_3$, $O_2$. |
| | Clouds | None |
| | Runcode | `PT7` |
| | Num. of runs | 1 |
| 8 | Name | WMGHG variation, M-star spectrum |
| | Description | As experiment 2, M-star spectrum substituted for solar spectrum. |
| | Spectrum | M-star |
| | Profile | GAM |
| | WHGHG | (a) $CO_2$ from $10^{-9}$ to $10^{-1}$, $1 \times 10^{-6}$ $CH_4$, $1 \times 10^{-6}$ $N_2O$. |
| | | (b) $400 \times 10^{-6}$ $CO_2$, $CH_4$ from $10^{-9}$ to $10^{-2}$, $1 \times 10^{-6}$ $N_2O$. |
| | | (c) $400 \times 10^{-6}$ $CO_2$, $1 \times 10^{-6}$ $CH_4$, $N_2O$ from $10^{-9}$ to $10^{-2}$. |
| | Absorbers | $CO_2$, $CH_4$, $N_2O$, $H_2O$, $O_3$, $O_2$. |
| | Clouds | None |
| | Runcode | `PT8a_`*x*, `PT8b_`*x*, `PT8c_`*x* for $CO_2$, $CH_4$ and $N_2O$ respectively. |
| | Num. of runs | $17 + 15 + 15 = 47$ |
| 9 | Name | WMGHG variation, high background, anoxic, M-star spectrum |

| Expt # | Parameter | Value/Desription |
|---|---|---|
| | Description | As experiment 3, M-star spectrum substituted for solar spectrum. |
| | Spectrum | M-star |
| | Profile | GAM |
| | WHGHG | (a) $CO_2$ from $10^{-9}$ to $10^{-1}$, $300 \times 10^{-6}$ $CH_4$, $30 \times 10^{-6}$ $N_2O$. |
| | | (b) $30,000 \times 10^{-6}$ $CO_2$, $CH_4$ from $10^{-9}$ to $10^{-2}$, $30 \times 10^{-6}$ $N_2O$. |
| | | (c) $30,000 \times 10^{-6}$ $CO_2$, $300 \times 10^{-6}$ $CH_4$, $N_2O$ from $10^{-9}$ to $10^{-2}$. |
| | Absorbers | $CO_2$, $CH_4$, $N_2O$, $H_2O$. |
| | Clouds | None |
| | Runcode | `PT9a_x`, `PT9b_x`, `PT9c_x` for $CO_2$, $CH_4$ and $N_2O$ respectively. |
| | Num. of runs | $17 + 15 + 15 = 47$ |
| 10 | Name | Water vapour variation, M-star spectrum |
| | Description | As experiment 3, M-star spectrum substituted for solar spectrum. |
| | Spectrum | M-star |
| | Profile | GAM, altered water vapour profiles |
| | WHGHG | $400 \times 10^{-6}$ $CO_2$, $1 \times 10^{-6}$ $CH_4$, $1 \times 10^{-6}$ $N_2O$. |
| | Absorbers | $CO_2$, $CH_4$, $N_2O$, $H_2O$, $O_3$, $O_2$. |
| | Clouds | None |
| | Runcode | `PT10_x` |
| | Num. of runs | 13 |
| 11 | Name | Surface pressure variation, M-star spectrum |
| | Description | As experiment 5, M-star spectrum substituted for solar spectrum. |
| | Spectrum | M-star |
| | Profile | GAM with modified pressure. |
| | WHGHG | $400 \times 10^{-6}$ $CO_2$, $1 \times 10^{-6}$ $CH_4$, $1 \times 10^{-6}$ $N_2O$. |
| | Absorbers | $CO_2$, $CH_4$, $N_2O$, $H_2O$. |
| | Clouds | None |
| | Runcode | `PT11_x` |
| | Num. of runs | 9 |
| 12 | Name | No oxygen or ozone absorption, M-star spectrum |
| | Description | As experiment 4, M-star spectrum substituted for solar spectrum. |
| | Spectrum | M-star |
| | Profile | GAM |

| Expt # | Parameter | Value/Desription |
|---|---|---|
| | WHGHG | $400 \times 10^{-6}$ $CO_2$, $1 \times 10^{-6}$ $CH_4$, $1 \times 10^{-6}$ $N_2O$. |
| | Absorbers | $CO_2$, $CH_4$, $N_2O$, $H_2O$. |
| | Clouds | None |
| | Runcode | PT12 |
| | Num. of runs | 1 |
| 13 | Name | Low cloud, thickness variation |
| | Description | A low altitude water cloud is added to the standard profile (experiment 1), and the liquid water path varied between 10 and 100 $g\,m^{-2}$. |
| | Spectrum | Solar |
| | Profile | GAM |
| | WHGHG | $400 \times 10^{-6}$ $CO_2$, $1 \times 10^{-6}$ $CH_4$, $1 \times 10^{-6}$ $N_2O$. |
| | Absorbers | $CO_2$, $CH_4$, $N_2O$, $H_2O$, $O_3$, $O_2$. |
| | Clouds | Water cloud, effective radius 10 $\mu$m, water path {10, 15, 25, 40, 63, 100} $g\,m^{-2}$. |
| | Runcode | PT13_*x* |
| | Num. of runs | 6 |
| 14 | Name | Low cloud, effective radius variation |
| | Description | A low altitude water cloud is added to the standard profile (experiment 1), and the effective radius varied between 5 and 25 $\mu$m. |
| | Spectrum | Solar |
| | Profile | GAM |
| | WHGHG | $400 \times 10^{-6}$ $CO_2$, $1 \times 10^{-6}$ $CH_4$, $1 \times 10^{-6}$ $N_2O$. |
| | Absorbers | $CO_2$, $CH_4$, $N_2O$, $H_2O$, $O_3$, $O_2$ |
| | Clouds | Water cloud, effective radius {5, 7.5, 10, 12.5, 15, 20, 25} $\mu$m, water path 40 $g\,m^{-2}$. |
| | Runcode | PT14_*x* |
| | Num. of runs | 7 |
| 15 | Name | High cloud, thickness variation |
| | Description | A high altitude water cloud is added to the standard profile (experiment 1), and the ice water path varied between 10 and 100 $g\,m^{-2}$. |
| | Spectrum | Solar |
| | Profile | GAM |
| | WHGHG | $400 \times 10^{-6}$ $CO_2$, $1 \times 10^{-6}$ $CH_4$, $1 \times 10^{-6}$ $N_2O$. |
| | Absorbers | $CO_2$, $CH_4$, $N_2O$, $H_2O$, $O_3$, $O_2$. |

| Expt # | Parameter | Value/Desription |
|---|---|---|
| | Clouds | Ice cloud, effective diameter 80 $\mu$m, water path {10, 15, 25, 40, 63, 100} g m$^{-2}$ |
| | Runcode | `PT15_`*x* |
| | Num. of runs | 6 |
| 16 | Name | High cloud, effective radius variation |
| | Description | A high altitude water cloud is added to the standard profile (experiment 1), and the effective diameter varied between 20 and 120 $\mu$m. |
| | Spectrum | Solar |
| | Profile | GAM |
| | WHGHG | $400 \times 10^{-6}$ CO$_2$, $1 \times 10^{-6}$ CH$_4$, $1 \times 10^{-6}$ N$_2$O. |
| | Absorbers | CO$_2$, CH$_4$, N$_2$O, H$_2$O, O$_3$, O$_2$. |
| | Clouds | Water cloud, effective diameter {20, 40, 60, 80, 100, 120} $\mu$m, water path 25 g m$^{-2}$ |
| | Runcode | `PT16_`*x* |
| | Num. of runs | 6 |

## 2.4 Submission of results

To facilitate comparison of many codes, each of which undoubtedly has its own output format, we ask that contributing scientists reformat output into the standard plain text format described below. These formats simple, and we have provided MATLAB codes which will write them automatically. These scripts, and sample output files, are available at www.palaeotrip.
5 org and included in the supplementary information for this paper.

For spectrally integrated output (dimensions W m$^{-2}$) the PALAEOTRIP data format consists of a plain text file with a twelve line header that includes the metadata in Table 3 followed by the data header describing each column, consisting the variables in Table 2. Each data column is twelve characters long. The formatting codes accept model output that corresponds to either pressure levels or layers and will automatically distinguish between these (levels are the boundary between model
10 layers). Quantities on layers and levels will be exported to separate data files but in both cases the first column will correspond to the pressure at the level or centre of the layer in pascals. The filename convention is `runcode_levels.txt` and `runcode_layers.txt` (e.g. `PT2a_1_layers.txt`,
`PT2a_1_layers.txt`).

For spectrally resolved output other than from line-by-line models, where available, a separate file should be provided for
15 each flux, with pressure levels as rows and each spectral bin as a column. There should be a twelve line header that includes the metadata in Table 3 and a field with the flux name, then column headers of the spectral bin edges in microns and the dimension of the column. The bin edges should be those native to the model. The fluxes in each bin should be provided in W m$^{-2}$ (that is the integrated flux within that spectral bin). If layer properties are provided, they likewise should be integrated

**Table 2.** Model output that will be accepted by PALAEOTRIP.

| Variable | Description | Unit |
|---|---|---|
| **Quantities on levels (bold variables are required):** | | |
| **plevel** | pressure on levels (layer bountaries) | Pa |
| Fswdndir | direct solar flux down | $W\,m^{-2}$ |
| Fswdndif | diffuse solar flux down | $W\,m^{-2}$ |
| Fswdn | total solar flux down (Fswdndir+Fswdndif) | $W\,m^{-2}$ |
| Fswup | solar flux up | $W\,m^{-2}$ |
| **Fswnet** | net solar flux | $W\,m^{-2}$ |
| Flwdn | thermal flux down | $W\,m^{-2}$ |
| Flwup | thermal flux up | $W\,m^{-2}$ |
| **Flwnet** | net thermal flux (Flwdn-Flwup) | $W\,m^{-2}$ |
| **Quantities on layers (all should be included if any are)** | | |
| player | pressure at layer centre | Pa |
| Qsolar | solar heating rate | $K\,day^{-1}$ |
| Qtherm | thermal heating rate | $K\,day^{-1}$ |

**Table 3.** Model metadata to be included with PALAEOTRIP submissions.

| Variable | Metadata Description |
|---|---|
| runcode | String with the code of run (see experiment descriptions) |
| modelname | String with the name (and version number) of model |
| username | String with your name (e.g. 'Colin Goldblatt') |
| useremail | String with your email (e.g. 'czg@uvic.ca') |
| usernotes | String with any notes about this run |

within each bin such that heating rates are in K/day for each bin. The filename convention is `runcode_variable.txt` (e.g. `PT2a_1_Fswdn.txt`).

All model output should be put into a single `.zip` file called `yourname_model.zip` and can be uploaded via the palaeotrip website. Include a `readme.txt` file as necessary.

5    For line-by-line models, spectrally resolved output should be subsampled to $1\,cm^{-1}$ resolution. Contact the PALAEOTRIP project team directly (info@palaeotrip.org) to discus how to submit this, as it will likely have too large a file size for our online submission system.

**Table 4.** Proposed PALAEOTRIP timeline

| Timeframe | Activity |
|---|---|
| January 2017 | Submit description/protocol paper |
| January – June 2017 | Review of description/protocol paper. Community feedback on experimental design. |
| May – June 2017 | Respond to review of protocol paper and finalize protocol. |
| July – August 2017 | Final protocol published |
| August – December 2017 | Contribution of radiative transfer model runs. |
| January – April 2018 | Nag participants for contributions. |
| May – July 2018 | Analysis of model output by PALAEOTRIP team. |
| July – August 2018 | Write results paper, circulate to co-authors. |
| September 2018 | Co-author comments. |
| October 2018 | Revise and submit results paper. |

## 3 Protocol and information for contributors

The final experimental design and protocols for the PALAEOTRIP are described in this paper. These were revised following formal review and informal discussion during the *Discussion* phase of the manuscript. If you intend to submit model output to the PALAEOTRIP project, we ask that you register your intention at www.palaeotrip.org or contact us directly. This will ensure that models are not run in duplicate by different groups, and that your model output is expected.

The anticipated timeline of the project is in Table 4. Sadly, few deadlines survive contact with academics, but we hope that this schedule is realistic and it is our intention to keep to it. We will post any updates to www.palaeotrip.org and communicate schedule changes directly to all participating scientists.

We intend that everyone submitting unique model results will be offered authorship on the final paper. Lead authorship will be by one of the project team, who will additionally determine the order of authorship (likely project team followed by contributing scientists, listed alphabetically). This paper will be circulated amongst all co-authors prior to submission.

A motivation of this project is to find out how a variety of radiation codes perform across a range of conditions applicable to palaeoclimate and exoplanets, so that future model users may know the range of conditions across which each model is likely to be accurate. Therefore, it is essential that models are able to be identified in the final paper. The analysis will be restricted to the range of conditions specified here, as an indicator of performance in palaeo- and exoclimate studies. We have no interest in, or intention of, commenting on the fitness of any model for any other purpose. It is the responsibility of scientist submitting model results to assert that the model can be identified in the final paper.

## 4 Summary and Discussion

PALAEOTRIP will run fourteen controlled experiments addressing the radiative transfer through a subset of conditions expected through Earth's past climate, and applicable to Earth-like exoplanets. We invite community participation in the exper-

iment. Over the course of the next year, the model runs will be performed and compared. The anticipated outcome is that the community will be better informed about the performance of available radiative transfer codes for palaeo- and exoclimate research.

The range of conditions which we have specified experiments for is somewhat "vanilla". It likely does not represent the full range of conditions seen in Earth's past, and will be a tiny fraction of the parameter space for Earth-like exoplanets. This is motivated to get wide participation; that is to specify conditions which most models which derive from Earth atmospheric sciences should be capable of being run for. We anticipate that, if this intercomparison is successful, we may be able to lead a more wide-ranging intercomparison in the future.

## 5 Code and data availability

A zip file containing the GAM profile, scripts to be used to write model output into the specified format and sample output is available in an online supplement to this article. Version corresponds to this manuscript. Updated versions will be made available through the project website, www.palaeotrip.org, as necessary.

Final model output will be available from www.palaeotrip.org and as an online supplement to the paper which will describe the results of the intercomparison.

*Author contributions.* CG has designed the experiment and is responsible for the scientific content herein. LK and MD have helped prepare materials and provided technical assistance.

*Competing interests.* The authors declare that they have no conflict of interest.

*Acknowledgements.* Thanks to Tony del Genio, Christos Matsoukas, Robert Pincus, Robin Wordsworth, Yun Yang and two anonymous reviewers for discussion and comments on the manuscript and draft protocol. Financial support has been provided by a NSERC discovery grant and UVic startup funds to CG.

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
