# Peer review of "The Palaeoclimate and Terrestrial Exoplanet Radiative Transfer Model Intercomparison Project (PALAEOTRIP): experimental design and protocols"

_Geoscientific Model Development, 2017_

## Referee Comment (RC1) · Anonymous Referee #1 · 6 Apr 2017

This is a well written paper describing the protocol for an intercomparison between radiation codes which will be useful for the Palaeoclimate and Terrestrial Exoplanet modelling communities.

The paper addresses a relevant scientific modelling question within the scope of GMD ; namely how to assess the skill of radiation codes used for a wide range of conditions which may be outside of those for which they were originally developed. It presents a modelling protocol that is suitable for addressing this question, involving the submission of outputs from radiative transfer codes run with standardised inputs covering a

range of conditions. These results will be compared with reference calculations from 'line-by-line' codes and published in a subsequent paper. The concepts involved are not particularly novel, being an extension of the approach used in previous studies; nonetheless they are useful because they will be applied to a wider range of radiation codes and conditions than previously. The results from this intercomparison will help modellers in the community to select the radiation codes best suited to their purposes, and to improve others, and so will be likely to result in substantial advances in modelling science.

The methods and assumptions are valid and clearly outlined, but unfortunately the description is not sufficiently complete and precise to allow the protocol to be executed by a modelling group.

I think the paper would be suitable for publication once the major issues below are addressed. I also list some minor issues which would be good to address.

Major issues:

1/ Page 9 I think that your timetable is very unrealistic. The amount of time you're giving people to send in their results is so small that there is a real risk of not getting sufficient participation to maximise value of this activity to the community. I would recommend seeking advice from other similar projects and coming up with something more realistic. I would have thought the modellers would need at least 6 months to send their data, and it would be a good idea to allow additional time for you to spot any errors in the data or its formatting and to allow them to resubmit.

2/ The supplementary information needs to be improved. The text of the paper reads "We have provided MATLAB and Python codes which will write them automatically from your output. These scripts, and sample output files, are available at www.palaeotrip.org and included in the supplementary information for this paper." In spite of what the text says, the SI does not contain any sample output files. These are needed because the text doesn't explain the full naming convention. These aren't available on the website

either at the time of writing, but even if they were, the protocol is supposed to be fully described by the paper. Also, the list of input files in incomplete ; for example there is no specification for the Stellar Spectra in the SI or on the website. Please also list the files individually in the readme file and say what each of them are. For example palaeotrip_profiles.mat file seems to be a binary and I have no idea what it is.

Minor issues:

1/ The abstract provides a concise and complete summary. A minor point however is that I found the aim to "constrain the ranges of far-from-modern atmospheric compositions in which the codes perform well" a bit unclear. Why would you want to constrain the ranges over which the codes perform well? I can understand why you would want to identify those ranges, and subsequently to allow to community to expand those ranges. Constraining them makes little sense to me.

2/ The overall presentation well generally well structured and clear. I did however find the list of experiments on page 5 somewhat redundant given the more informative and detailed list which appears on Page 6. I would recommend merging the list on Page 6 into the list on Page 5. If you want an 'at a glance' summary of the experiments then I would add a table.

3/ Page 1 Line 1 time on the order of

4/ Page 3 Line 25. It might be nice to explain why a single global mean profile is considered sufficient. Previous intercomparisons have used profiles from different regimes/seasons, e.g. McClatchey Mid-Latitude Summer/Winter etc.

5/ Page 3 Line 31 number/name, a brief

6/ Page 5 Line 9 is expected

7/ Page 6. Please be consistent - i.e. Experiment 1 / Experiments 11.

9/ Page 7 Line 2 which is best done

10/ Page 7 Line 8 with a

11/ Page 7 Line 9 consisting of the

12/ Table 1 diffuse

---

## Referee Comment (RC2) · Anonymous Referee #2 · 6 Apr 2017

Review of gmd-2017-24 "The Palaeoclimate and Terrestrial Exoplanet Radiative Transfer Model Intercomparison Project (PALAEOTRIP): experimental design and protocols"

The manuscript introduces a new intercomparison project for radiation codes, PALAEOTRIP, aimed at evaluating the accuracy of and differences between radiation codes applied to study paleo- and terrestrial exoplanet climates. As the field of modelling climates of planets significantly different from the present-day Earth matures, it will be increasingly important to evaluate the accuracy of the radiation schemes used, as radiative transfer is one of the most important components of climate models. I

commend the authors for taking the initiative to begin such an intercomparison.

I found the manuscript to be well written, and the different experiments to be explained in sufficient detail. My main concerns are with the large number of different runs proposed (> 200), that important parts of the parameter space are not included, and that the experiments including clouds will lead to differences between codes that may not be errors and have no distinction from conditions found on present day Earth. I discuss my concerns in more detail below, and recommend publication once my major concerns have been addressed.

To facilitate broad participation, the authors could consider adopting an experiment design similar to that proposed for CMIP6, with a core set of experiments that all participating groups are expected to do, and with remaining experiments organised in terms of increasing optionality: http://www.mpimet.mpg.de/en/communication/news/single-news/?no_cache=1&tx_ttnews%5Btt_news%5D=606

Main comments:

1) Experiment 2: In this experiment, well-mixed greenhouse gas (WMGHG) amounts are varied. Only one gas is varied at a time, with the rest kept at standard conditions, amounting to a total of 113 different runs. I have several suggestions on how this experiment could be improved:

1.1) I think the number of experiments here is unnecessarily large, which may put off some potential participants. I think the number of gas concentrations per log unit can be reduced to one or two without losing a significant amount of information.

1.2) The maximum N2O amount seems quite large to me, I am curious about how the authors arrived at this number (1e-2 volume mixing ratio). Also, some radiation schemes may not include N2O, so it might be worth having some experiments without N2O to facilitate broader participation.

1.3) As only one gas is varied at any given time, a significant part of the parameter

space is not being considered, including the authors' example of a typical late Archean atmospheric composition in the introduction where both CO2 and CH4 amounts are elevated compared to present day Earth. I think it would be beneficial to add some runs with compositions that have previously been used in climate models of the Archean Earth.

1.4) Should these experiments include oxygen and ozone? The Archean atmosphere is thought to have had very little oxygen and ozone, including some experiments without these absorbers may be useful.

In summary, I would encourage the authors to significantly reduce the number of WMGHG amounts in this experiment and also to include other, very common compositions such as atmospheres where both CO2 and CH4 amounts are elevated compared to present day Earth.

2) Experiment 3: The water vapour mixing ratio could be reduced even further in this experiment, perhaps by a factor of 0.01 or 0.001. Five mixing ratios per log unit may also be unnecessarily many, this could potentially be reduced to two or three. Also, planets receiving large near-IR fluxes may have very large stratospheric water vapour mixing ratios (up to ∼1e-3), the authors could consider adding an experiment with a modified water vapour profile where the stratospheric water vapour amount is elevated compared to present day Earth.

3) Experiment 5: Why have the authors decided to turn off oxygen absorption while still having ozone absorption turned on? This test also moves the upper boundary to a higher or lower pressure. In particular, the largest surface pressure will move the upper boundary to 1 mbar, which is a rather large pressure. I would recommend defining a few separate P-T profiles with varying surface pressures (but constant upper boundary pressures) to use for this test instead of simply multiplying the values in the GAM profile with a constant factor.

4) Experiments 6-8: WMGHG amounts are not varied for the experiments using an

M-star spectrum. $CO_2$, and particularly $CH_4$, are significant near-IR absorbers. It would be very interesting to see how well the different radiation schemes deal with the overlapping absorption in the near-IR between $H_2O$, $CO_2$ and $CH_4$ for cases with large amounts of $CO_2$ and $CH_4$. Any errors in this region can become significantly larger with an M-star spectrum compared to that obtained with a Sun-like star spectrum due to the large near-IR flux.

5) The temperature-pressure profile is kept the same in all experiments (except in experiment 5 where it is scaled to achieve a smaller/larger surface pressure). For the stellar (short-wave) component of the radiation I would not expect errors in most radiation codes to depend strongly on temperature (except if temperatures become high enough to warrant the use of high temperature line lists). Errors in the thermal (long-wave) radiation, however, can depend on the temperature due to the shift in the peak of the Planck function with temperature, which will emphasise different wavelengths. It may be worth adding another experiment where the temperature is varied within a reasonable range to see how well codes deal with somewhat lower and higher temperatures than those found on present day Earth.

6) Experiments 9-12 involve adding a low or high altitude cloud to the setup of experiment 1 and vary the water path or cloud particle size. Currently these experiments feel somewhat out-of-place:

6.1) The motivation for including these experiments is not clear from the current manuscript. The other experiments are designed to test how well radiation codes perform for conditions potentially significantly different from present day Earth. In these experiments conditions are similar to those found on present day Earth, and most approximations used have been tested for these conditions by present day Earth climate modellers (see e.g. Oreopoulos et al. 2012, http://dx.doi.org/10.1029/2011JD016821, Barker et al. 2015, http://dx.doi.org/10.1175/JAS-D-15-0033.1). I think a stronger motivation for these experiments is required.

6.2) Experiments 11-12 include ice clouds with a prescribed effective size D_eff with optical properties from Baum et al. (2014). For several participating groups this may involve implementing new ice cloud scattering properties in their radiation codes, solely for the purpose of participating in this intercomparison. I think it may be too much to ask groups to do this, and results would not directly reflect those obtained in the respective climate models.

6.3) It is not clear how the benchmark results will be defined and obtained in these tests. Different and entirely reasonable choices with regards to e.g. the size distribution of cloud particles may result in differences between radiation codes that cannot be considered to be errors as in the other experiments. This should be discussed in more detail.

6.4) The number of different runs may also here be unnecessarily large, 54 in total. I would suggest reducing it to about three runs per experiment (e.g. with a low, medium and high value) to ease participation.

I my opinion these points will need to be addressed in order to justify including Experiments 9-12 in this intercomparison.

Minor comments:

7) The abstract and introduction paints a rather negative view of the current state of radiation codes used to study paleo- and terrestrial exoplanet climates. While it is true that the accuracy of several radiation codes remains unevaluated, at least in the literature, there have been some work to address this. Examples are Wolf & Toon (2013) (dx.doi.org/10.1089/ast.2012.0936), who evaluated the accuracy of their new radiation scheme by comparing it to the LBLRTM, and Yang et al. (2016) (dx.doi.org/10.3847/0004-637X/826/2/222), who evaluated differences between several radiation schemes when applied to the inner edge of the habitable zone. These works should be mentioned and referenced.

[Figure]

8) Introduction, first paragraph, first sentence ("A typical model of ..."): One or more references are needed. Also, giving gas amounts in units of pressure is ambiguous (a gas' contribution to the surface pressure and the gas' partial pressure at the surface are generally different). Please consider using ppmv for all gas amount units, or clarify which pressure is used.

9) Introduction, second paragraph: The statement that deriving the surface temperature for a given atmospheric composition and incident flux is conceptually a simple physics problem is somewhat oversimplifying the problem. Uncertainties in e.g. ground albedos, cloud physics and ocean heat transport (with a potentially unknown land/ocean distribution) can potentially impact surface temperatures significantly. In my opinion this discussion should be modified to argue for why performing accurate radiative transfer is both important and difficult, while at the same time acknowledging that other uncertainties remain.

10) Introduction, second paragraph: In my experience line-by-line calculations can, with a reasonable number of layers ($\sim$ 40), take as little as a few minutes for a single column. Still several orders of magnitude too slow for use in a GCM, but not as bad as indicated.

11) Introduction, third paragraph: A statement is made that atmospheric composition is equivalent to column abundances. This is strictly speaking not correct as gas mixing ratios are 3D fields, while column densities are vertically integrated fields.

12) Section 2.1, second paragraph: To argue for why H2-dominated atmospheres are not included, it is stated that the altered mean molar weight would lead to different pressure-broadened line shapes. While it is indeed true that H2 pressure-broadened widths are different from air-broadened widths, this is not only due to H2 molecules being lighter than air molecules; calculating pressure-broadened line widths is a rather complicated quantum-mechanical problem. Please reformulate.

13) Section 2.2.1: I assume the mixing ratios provided online with the GAM profile are

volume mixing ratios, but I could not find this specified anywhere. Also, it would be nice if the GAM profile could be specified on both levels and layers to avoid potential slight inconsistencies between codes.

14) Section 2.2.3: Will the supplied stellar spectra be normalised such that, integrated over wavelength, they give the TOA flux to be used in the experiments? Otherwise the TOA flux will need to be specified.

15) Section 2.2.5: The effective temperature of the surface is missing.

16) Section 2.3: Currently, the list of experiments is provided twice, one on the form of an overview and one as a list with details on each experiment. I understand why, but to me this seems a bit awkward. I would consider making a large table with details on the different experiments to provide a better overview, and refer to this in the main text when discussing them.

17) Section 2.4, second paragraph: Consider moving the definition of layers and levels to section 2.2.4 as they are used there.

18) Please consider adding more references to recent radiation intercomparisons, e.g.: Oreopoulos et al. (2012): http://dx.doi.org/10.1029/2011JD016821 Pincus et al. (2015): http://dx.doi.org/10.1002/2015GL064291

19) From statements in section 2.2.2, and 2.4, I deduce that benchmark results from line-by-line codes are meant to be submitted along with results from other radiation codes. Please make this more clear.

Typos:

- Page 3, line 31: "a a" –> "a"

- Page 4, line 4: "and and" –> "and an"

- Page 6, line 22: Runcode for experiment 8 should be PT8_x.

- Page 7, lines 8-9: "an ten line" –> "a ten line"
* * *

---

## Author Comment (AC1) · 22 Jun 2017

We thank Reviewer 1 for a helpful review of our experimental protocol. We provide a full response to the reviewers comments (reproduced in italics) below, together with revisions to the manuscript.

*This is a well written paper describing the protocol for an intercomparison between radiation codes which will be useful for the Palaeoclimate and Terrestrial Exoplanet modelling communities. The paper addresses a relevant scientific modelling question*

[Figure]

*within the scope of GMD ; namely how to assess the skill of radiation codes used for a wide range of conditions which may be outside of those for which they were originally developed. It presents a modelling protocol that is suitable for addressing this question, involving the submission of outputs from radiative transfer codes run with standardised inputs covering a range of conditions. These results will be compared with reference calculations from 'line-by-line' codes and published in a subsequent paper. The concepts involved are not particularly novel, being an extension of the approach used in previous studies; nonetheless they are useful because they will be applied to a wider range of radiation codes and conditions than previously. The results from this intercomparison will help modellers in the community to select the radiation codes best suited to their purposes, and to improve others, and so will be likely to result in substantial advances in modelling science. The methods and assumptions are valid and clearly outlined, but unfortunately the description is not sufficiently complete and precise to allow the protocol to be executed by a modelling group. I think the paper would be suitable for publication once the major issues below are addressed. I also list some minor issues which would be good to address.*

We address specific issues discussed raised here below.

**Major issues:**
*1/ Page 9 I think that your timetable is very unrealistic. The amount of time you're giving people to send in their results is so small that there is a real risk of not getting sufficient participation to maximise value of this activity to the community. I would recommend seeking advice from other similar projects and coming up with something more realistic. I would have thought the modellers would need at least 6 months to send their data, and it would be a good idea to allow additional time for you to spot any errors in the data or its formatting and to allow them to resubmit.*

Sober reflection and plain passage of time has us agree on this point. We have re-laxed the timescale very substantially, with the goal of a summer 2018 completion, not summer 2017.

*2/ The supplementary information needs to be improved. The text of the paper reads "We have provided MATLAB and Python codes which will write them automatically from your output. These scripts, and sample output files, are available at www.palaeotrip.org and included in the supplementary information for this paper." In spite of what the text says, the SI does not contain any sample output files. These are needed because the text doesn't explain the full naming convention. These aren't available on the website either at the time of writing, but even if they were, the protocol is supposed to be fully described by the paper. Also, the list of input files in incomplete; for example there is no specification for the Stellar Spectra in the SI or on the website. Please also list the files individually in the readme file and say what each of them are. For example palaeotrip_profiles.mat file seems to be a binary and I have no idea what it is.*

The SI is revised, completed, and fully described in a readme file.

***Minor issues:***
*1/ The abstract provides a concise and complete summary. A minor point however is that I found the aim to "constrain the ranges of far-from-modern atmospheric compositions in which the codes perform well" a bit unclear. Why would you want to constrain the ranges over which the codes perform well? I can understand why you would want to identify those ranges, and subsequently to allow to community to expand those ranges. Constraining them makes little sense to me.*

We have changed "constrain" to "identify".

*2/ The overall presentation well generally well structured and clear. I did however find the list of experiments on page 5 somewhat redundant given the more informative and detailed list which appears on Page 6. I would recommend merging the list on Page 6 into the list on Page 5. If you want an ?at a glance? summary of the experiments then I would add a table.*

The descriptions have been merged into a new table, Table 1.

*3/ Page 1 Line 1 time on the order of*
Fixed.

*4/ Page 3 Line 25. It might be nice to explain why a single global mean profile is considered sufficient. Previous intercomparisons have used profiles from different regimes/seasons, e.g. McClatchey Mid-Latitude Summer/Winter etc.*

We now say "For simplicity, all experiments use a Global Annual Mean (GAM) profile". Others would be nice, but this simple approach is probably sufficient. There isn't much more to say!

*5/ Page 3 Line 31 number/name, a brief*
*6/ Page 5 Line 9 is expected*
*7/ Page 6. Please be consistent - i.e. Experiment 1 / Experiments 11.*
*9/ Page 7 Line 2 which is best done*
*10/ Page 7 Line 8 with a*
*11/ Page 7 Line 9 consisting of the*
*12/ Table 1 diffuse*

All fixed.

---

## Author Comment (AC2) · 22 Jun 2017

We thank Reviewer 2 for an exceptionally detailed review of our experimental protocol. We provide a full response to the reviewers comments (reproduced in italics) below, together with revisions to the manuscript.

*The manuscript introduces a new intercomparison project for radiation codes, PALAEOTRIP, aimed at evaluating the accuracy of and differences between radiation codes applied to study paleo- and terrestrial exoplanet climates. As the field of mod-*

[Figure]
*elling climates of planets significantly different from the present-day Earth matures, it will be increasingly important to evaluate the accuracy of the radiation schemes used, as radiative transfer is one of the most important components of climate models. I commend the authors for taking the initiative to begin such an intercomparison.*

Thank you! We hope this project will be useful to the community.

*I found the manuscript to be well written, and the different experiments to be explained in sufficient detail. My main concerns are with the large number of different runs proposed (> 200), that important parts of the parameter space are not included, and that the experiments including clouds will lead to differences between codes that may not be errors and have no distinction from conditions found on present day Earth. I discuss my concerns in more detail below, and recommend publication once my major concerns have been addressed.*

We address these points in more detail below. In summary...

*To facilitate broad participation, the authors could consider adopting an experiment design similar to that proposed for CMIP6, with a core set of experiments that all participating groups are expected to do, and with remaining experiments organised in terms of increasing optionality: http://www.mpimet.mpg.de/en/communication/news/single-news/?no_cache=1&tx_ttnews%5Btt_news%5D=606.*

In section 2.1, we now say:
"Participating groups should run the experiments that their models are configured for, and omit any which are not possible (or onerous) to run. We do not expect groups do perform model development in order to participate in this project. For example, a model which had the solar spectrum hard coded and did not include N2O absorption would run experiments 1, 2a–b, 3–6 and 13–16. A model without clouds would omit experiments 13–16. If, for any reason, there is a limit to the number of experiments that a group can run then experiments 1–6 should be considered "core" and prioritized. A minimal set of experiments would be 1 and 2. "

*Main comments:*

*1) Experiment 2: In this experiment, well-mixed greenhouse gas (WMGHG) amounts are varied. Only one gas is varied at a time, with the rest kept at standard conditions, amounting to a total of 113 different runs. I have several suggestions on how this experiment could be improved:*

*1.1) I think the number of experiments here is unnecessarily large, which may put off some potential participants. I think the number of gas concentrations per log unit can be reduced to one or two without losing a significant amount of information.*

We have reduced the number to two per log-unit. This makes the lead author a bit jittery to have such course spacing, but I accept it is probably for the best.

*1.2) The maximum N2O amount seems quite large to me, I am curious about how the authors arrived at this number (1e-2 volume mixing ratio). Also, some radiation schemes may not include N2O, so it might be worth having some experiments without N2O to facilitate broader participation.*

We made this up. No-one has any idea of what Archean or Proterozoic N2O levels were (I say this as someone who works on the early nitrogen cycle!). Fluxes may well have been quite high given incomplete denitrification in suboxic environments. Even a first order estimate would be a good paper, but is beyond the scope here. We have set the upper bound high to be inclusive.

As we emphasize further now in section 2.1 (see above), groups should run experiments which they can, a model without N2O should run 2a and 2b, but not 2c. As N2O absorbtion would be omitted in the standard run as well as experiments 2 and higher, then comparison of differences between test and standard conditions would still be meaningful.

*1.3) As only one gas is varied at any given time, a significant part of the parameter space is not being considered, including the authors example of a typical late Archean*

*atmospheric composition in the introduction where both CO2 and CH4 amounts are elevated compared to present day Earth. I think it would be beneficial to add some runs with compositions that have previously been used in climate models of the Archean Earth.*

We have included a new experiment with overlap as experiment 3 for solar, and 9 for M-star. To keep the setup simple, this simply replaces standard background with a nominal set of high background levels.

*1.4) Should these experiments include oxygen and ozone? The Archean atmosphere is thought to have had very little oxygen and ozone, including some experiments without these absorbers may be useful.*

Indeed, the Archean atmosphere was around 1ppmv O2 and no O3, whereas Phanerozoic O2 and O3 levels are essentially modern. Thus there is a dilemma about what levels to use. For the purpose of an intercomparison, however, our focus in on simple experiments on gas addition, relative to standard conditions. Therefore, we have kept O2 and O3 in when changing most WHGHGs. There is the issue of overlap of course, but this should be minor: O3 absorption overlaps with CO2 only in the thermal region.

*In summary, I would encourage the authors to significantly reduce the number of WMGHG amounts in this experiment and also to include other, very common compositions such as atmospheres where both CO2 and CH4 amounts are elevated compared to present day Earth.*

*2) Experiment 3: The water vapour mixing ratio could be reduced even further in this experiment, perhaps by a factor of 0.01 or 0.001. Five mixing ratios per log unit may also be unnecessarily many, this could potentially be reduced to two or three. Also, planets receiving large near-IR fluxes may have very large stratospheric water vapour mixing ratios (up to 1e-3), the authors could consider adding an experiment with a modified water vapour profile where the stratospheric water vapour amount is elevated compared to present day Earth.*
* * *
Interactive
comment

We have reduced the minimum factor to 0.01.

A specific moist stratosphere experiment would be interesting, but this would be more complicated to set up and we are motivated (and advised by this reviewer!) to keep the number of experiments simple. The experiments with high water vapour with the M-star spectrum will offer some guidance here. Thus we do not wish to add an additional experiment.

*3) Experiment 5: Why have the authors decided to turn off oxygen absorption while still having ozone absorption turned on? This test also moves the upper boundary to a higher or lower pressure. In particular, the largest surface pressure will move the upper boundary to 1 mbar, which is a rather large pressure. I would recommend defining a few separate P-T profiles with varying surface pressures (but constant upper boundary pressures) to use for this test instead of simply multiplying the values in the GAM profile with a constant factor.*

Re GHG concentrations: the motivation in this experiment was to keep amounts of each absorber constant. For minor species, this is easy to achieve, as described. However, it is obviously impossible for oxygen when surface pressure is reduced strongly. Therefore, oxygen absorption is turned of in all for self-consistency. There is a minor loss of physical realism (though oxygen absorption is minor), which is justified because the motivation is self-consistent inter-comparison of codes, not accurate climate prediction.

Re upper boundary pressure, in the standard case the difference in flux between 0.1 and 1mbar is $< 0.1\,\mathrm{W\,m^{-2}}$ in all radiation streams (sample output now in SI), therefore we do not expect this to cause a problem.

*4) Experiments 6-8: WMGHG amounts are not varied for the experiments using an M-star spectrum. CO2, and particularly CH4, are significant near-IR absorbers. It would be very interesting to see how well the different radiation schemes deal with the overlapping absorption in the near-IR between H2O, CO2 and CH4 for cases with large*

*amounts of CO2 and CH4. Any errors in this region can become significantly larger with an M-star spectrum compared to that obtained with a Sun-like star spectrum due to the large near-IR flux.*

Fair point. So, we have added additional experiments to make the suite of experiments for an M-star spectrum identical to those for the solar spectrum. We considered further picking-and-choosing, but the simpler approach of duplicating all seemed easier (all participatiing groups should then have to do is change the spectrum for each, so all GHG experiments can be done for both spectra).

*5) The temperature-pressure profile is kept the same in all experiments (except in experiment 5 where it is scaled to achieve a smaller/larger surface pressure). For the stellar (short-wave) component of the radiation I would not expect errors in most radiation codes to depend strongly on temperature (except if temperatures become high enough to warrant the use of high temperature line lists). Errors in the thermal (long-wave) radiation, however, can depend on the temperature due to the shift in the peak of the Planck function with temperature, which will emphasise different wavelengths. It may be worth adding another experiment where the temperature is varied within a reasonable range to see how well codes deal with somewhat lower and higher temperatures than those found on present day Earth.*

This would be an interesting experiment. However, we have added other experiments in response to this review, and also have the mandate to keep the total number of experiments low.

*6) Experiments 9-12 involve adding a low or high altitude cloud to the setup of experiment 1 and vary the water path or cloud particle size. Currently these experiments feel somewhat out-of-place:*
*6.1) The motivation for including these experiments is not clear from the current manuscript. The other experiments are designed to test how well radiation codes perform for conditions potentially significantly different from present day Earth. In these*

*experiments conditions are similar to those found on present day Earth, and most approximations used have been tested for these conditions by present day Earth climate modellers (see e.g. Oreopoulos et al. 2012, http://dx.doi.org/10.1029/2011JD016821, Barker et al. 2015, http://dx.doi.org/10.1175/JAS-D-15-0033.1). I think a stronger motivation for these experiments is required.*

We now add to the manuscript:
"There a range of good choices of representation of cloud microphysics in models (i.e. which are different but entirely reasonable), so variation in the radiative effects of clouds may arise from these rather than error per se. Nonetheless, it is of primary interest to us how the radiative effects of clouds do vary when every attempt has been made to specify cloud physical properties equivalently."

*6.2) Experiments 11-12 include ice clouds with a prescribed effective size Deff with optical properties from Baum et al. (2014). For several participating groups this may involve implementing new ice cloud scattering properties in their radiation codes, solely for the purpose of participating in this intercomparison. I think it may be too much to ask groups to do this, and results would not directly reflect those obtained in the respective climate models.*

This is a misunderstanding of our intention, so we have improved the clarity of the manuscript. We add:
"We emphasize that the normal implementations of clouds in participant models should be used; single scattering properties are provided only for cases where this necessarily needs to be input. "

*6.3) It is not clear how the benchmark results will be defined and obtained in these tests. Different and entirely reasonable choices with regards to e.g. the size distribution of cloud particles may result in differences between radiation codes that cannot be considered to be errors as in the other experiments. This should be discussed in more detail.*

We now state in the manuscript:

" There a range of good choices of representation of cloud microphysics in models (i.e. which are different but entirely reasonable), so variation in the radiative effects of clouds may arise from these rather than error per se. Nonetheless, it is of primary interest to us how the radiative effects of clouds do vary when every attempt has been made to specify cloud physical properties equivalently."

*6.4) The number of different runs may also here be unnecessarily large, 54 in total. I would suggest reducing it to about three runs per experiment (e.g. with a low, medium and high value) to ease participation.*

We have reduced it to 6 or 7 runs per experiments, total 25. In our opinion, three would simply not be enough.

*I my opinion these points will need to be addressed in order to justify including Experiments 9-12 in this intercomparison.*

All the points are addressed above.

**Minor comments:**

*7) The abstract and introduction paints a rather negative view of the current state of radiation codes used to study paleo- and terrestrial exoplanet climates. While it is true that the accuracy of several radiation codes remains unevaluated, at least in the literature, there have been some work to address this. Examples are Wolf & Toon (2013) (dx.doi.org/10.1089/ast.2012.0936), who evaluated the accuracy of their new radiation scheme by comparing it to the LBLRTM, and Yang et al. (2016) (dx.doi.org/10.3847/0004-637X/826/2/222), who evaluated differences between several radiation schemes when applied to the inner edge of the habitable zone. These works should be mentioned and referenced.*

Now cited.

*8) Introduction, first paragraph, first sentence ("A typical model of ..."): One or more*

*references are needed. Also, giving gas amounts in units of pressure is ambiguous (a gas? contribution to the surface pressure and the gas? partial pressure at the surface are generally different). Please consider using ppmv for all gas amount units, or clarify which pressure is used.*

We've changed ppm to ppmv for clarity, and used percent for CO2, and changed "A typical model of" to "An example model of". There isn't one or a few good references for the nominal composition chosen, and it won't help maters to put in a few paragraphs of justification here - it is really just an example to set the tone.

*9) Introduction, second paragraph: The statement that deriving the surface temperature for a given atmospheric composition and incident flux is conceptually a simple physics problem is somewhat oversimplifying the problem. Uncertainties in e.g. ground albedos, cloud physics and ocean heat transport (with a potentially unknown land/ocean distribution) can potentially impact surface temperatures significantly. In my opinion this discussion should be modified to argue for why performing accurate radiative transfer is both important and difficult, while at the same time acknowledging that other uncertainties remain.*

We have added in "surface properties specified"; in respect to other points we beg to be allowed some artistic licence in motivating the experiment.

*10) Introduction, second paragraph: In my experience line-by-line calculations can, with a reasonable number of layers (? 40), take as little as a few minutes for a single column. Still several orders of magnitude too slow for use in a GCM, but not as bad as indicated.*

We now say "minutes to hours".

*1) Introduction, third paragraph: A statement is made that atmospheric composition is equivalent to column abundances. This is strictly speaking not correct as gas mixing ratios are 3D fields, while column densities are vertically integrated fields.*

[Figure]

We have revised this to remove the erroneous statement of equivalence: "To optimize efficiency, these parameterizations may made for limited ranges of atmospheric composition or column abundances of absorbing molecules."

*12) Section 2.1, second paragraph: To argue for why H2-dominated atmospheres are not included, it is stated that the altered mean molar weight would lead to different pressure-broadened line shapes. While it is indeed true that H2 pressure-broadened widths are different from air-broadened widths, this is not only due to H2 molecules being lighter than air molecules; calculating pressure-broadened line widths is a rather complicated quantum-mechanical problem. Please reformulate.*

We now say: "One class of model atmospheres that we exclude is H2 dominated atmospheres (Wordsworth, 2013), as air-broadened line shapes will likely not be appropriate and thus a majority of codes may not perform well (that is, these atmospheres require rather specialist treatment, beyond the scope of this intercomparison). "

*13) Section 2.2.1: I assume the mixing ratios provided online with the GAM profile are volume mixing ratios, but I could not find this specified anywhere. Also, it would be nice if the GAM profile could be specified on both levels and layers to avoid potential slight inconsistencies between codes.*

These are indeed volume mixing ratios. This is now specified at 2.2.1

We have additionally provided layers in the SI.

*14) Section 2.2.3: Will the supplied stellar spectra be normalised such that, integrated over wavelength, they give the TOA flux to be used in the experiments? Otherwise the TOA flux will need to be specified.*

Yes, and a solar constant is now additionally specified.

*15) Section 2.2.5: The effective temperature of the surface is missing.*

Now specified.

*16) Section 2.3: Currently, the list of experiments is provided twice, one on the form of an overview and one as a list with details on each experiment. I understand why, but to me this seems a bit awkward. I would consider making a large table with details on the different experiments to provide a better overview, and refer to this in the main text when discussing them.*

Fair point. We have moved all of this into "One Table to bring them all and in the lightness bind them".

*17) Section 2.4, second paragraph: Consider moving the definition of layers and levels to section 2.2.4 as they are used there.*

Done.

*18) Please consider adding more references to recent radiation intercomparisons, e.g.: Oreopoulos et al. (2012): http://dx.doi.org/10.1029/2011JD016821 Pincus et al. (2015): http://dx.doi.org/10.1002/2015GL064291*

These are now cited.

*9) From statements in section 2.2.2, and 2.4, I deduce that benchmark results from line-by-line codes are meant to be submitted along with results from other radiation codes. Please make this more clear.*

We state clearly:"For line-by-line models, spectrally resolved output should be subsampled to 1 cm?1 resolution. Contact the PALAEOTRIP project team directly to discus how to submit this (info@palaeotrip.org)." The point is that LBL output may be too large for the online submission system that we have.

*Typos*
*-Page3,line31: "aa" –"a"*
*- Page 4, line 4: "and and" – "and an"*
*- Page 6, line 22: Runcode for experiment 8 should be PT8_x.*
*- Page 7, lines 8-9: "an ten line" – "a ten line"*

---

## Author Response (AR2)

University of Victoria
PO Box 3065
Victoria  BC  V8W 3V6  Canada

Tel      (250) 721-6120
Fax      (250) 721-6200
Email    seos@uvic.ca
Web      www.seos.uvic.ca

**School of Earth and Ocean Sciences**

**Dr Colin Goldblatt**
*Associate Professor*
Tel      (250) 472 4060
Email   czg@uvic.ca

18th August 2017

Dear Dr Hargreaves,

I enclose the revised version of our manuscript. These changes are very minor, so I list them here:

Reviewer #1:

P2 L5 parameterizations may be made for
P2 L30 such a systematic intercomparison
Table 1 Paremeter -> Parameter

All these typos are corrected.

Reviewer #2:

I would like to thank the authors for addressing the majority of my concerns, particularly for reducing the number of runs and for adding experiments 3 and 9. I only have two minor concerns, once these have been addressed I will be happy to recommend the paper for publication in GMD.

1) On page 1, lines 14 to 19: The authors still state that deriving the surface temperature of a planet "is a conceptually simple physics problem". As I said in my previous report, this is not true, and the authors do not need to say this to motivate their intercomparison project. To motivate their study the authors should rather argue that performing accurate radiative transfer calculations is the number one priority in any climate model.

We have replaced "This is a conceptually simple physics problem" with "To first approximation".

2) I am concerned that radiation schemes designed for anoxic atmospheres may not be able to participate in most of these experiments. To avoid adding more experiments, I suggest that O2 and O3 could be removed from experiments 3 and 9. I believe this would both broaden and ease participation in these experiments, but I will leave the final decision on this to the authors.

This is a good idea, which we have adopted.

Additionally, we have added the following paragraph to section 2.1:
"If some absorbing gases are missing, experiments which do not focus on these can still be run, with notes in the metadata and in discussion with the project team. As or analysis will focus on forcings (change from standard conditions) comparison to the standard conditions *from that model* will minimise the effect of any systematic offset from missing absorbers. For example, models without oxygen or ozone absorption could still run the experiments focussing on clouds."

I have appended a graphical diff made with LaTeXdiff. Thank you again for considering this manuscript.

Yours sincerely,

Colin Goldblatt